# LRRC46 Accumulates at the Midpiece of Sperm Flagella and Is Essential for Spermiogenesis and Male Fertility in Mouse

**DOI:** 10.3390/ijms23158525

**Published:** 2022-07-31

**Authors:** Yingying Yin, Wenyu Mu, Xiaochen Yu, Ziqi Wang, Ke Xu, Xinyue Wu, Yuling Cai, Mingyu Zhang, Gang Lu, Wai-Yee Chan, Jinlong Ma, Tao Huang, Hongbin Liu

**Affiliations:** 1Center for Reproductive Medicine, Shandong University, Jinan 250012, China; 18865381921@163.com (Y.Y.); 17853137722@163.com (W.M.); yuxiaochen@mail.sdu.edu.cn (X.Y.); wangziqi1110@126.com (Z.W.); 202020745@mail.sdu.edu.cn (K.X.); wxy17853138133@163.com (X.W.); caiyuling37@163.com (Y.C.); zmyscience@163.com (M.Z.); majinlong123@hotmail.com (J.M.); 2Research Unit of Gametogenesis and Health of ART-Offspring, Chinese Academy of Medical Sciences, Jinan 250012, China; 3Key Laboratory of Reproductive Endocrinology of Ministry of Education, Shandong University, Jinan 250012, China; 4CUHK-SDU Joint Laboratory on Reproductive Genetics, School of Biomedical Sciences, The Chinese University of Hong Kong, Hong Kong, China; lugang@cuhk.edu.hk (G.L.); chanwy@cuhk.edu.hk (W.-Y.C.)

**Keywords:** multiple morphological abnormalities of the sperm flagella (MMAF), leucine rich repeat containing 46 (LRRC46), male infertility

## Abstract

The sperm flagellum is essential for male fertility. Multiple morphological abnormalities of the sperm flagella (MMAF) is a severe form of asthenoteratozoospermia. MMAF phenotypes are understood to result from pathogenic variants of genes from multiple families including AKAP, DANI, DNAH, RSPH, CCDC, CFAP, TTC, and LRRC, among others. The Leucine-rich repeat protein (LRRC) family includes two members reported to cause MMAF phenotypes: *Lrrc6* and *Lrrc50*. Despite vigorous research towards understanding the pathogenesis of MMAF-related diseases, many genes remain unknown underlying the flagellum biogenesis. Here, we found that Leucine-rich repeat containing 46 (LRRC46) is specifically expressed in the testes of adult mice, and show that LRRC46 is essential for sperm flagellum biogenesis. Both scanning electron microscopy (SEM) and Papanicolaou staining (PS) presents that the knockout of *Lrrc46* in mice resulted in typical MMAF phenotypes, including sperm with short, coiled, and irregular flagella. The male KO mice had reduced total sperm counts, impaired sperm motility, and were completely infertile. No reproductive phenotypes were detected in *Lrrc46^−/−^* female mice. Immunofluorescence (IF) assays showed that LRRC46 was present throughout the entire flagella of control sperm, albeit with evident concentration at the mid-piece. Transmission electron microscopy (TEM) demonstrated striking flagellar defects with axonemal and mitochondrial sheath malformations. About the important part of the Materials and Methods, SEM and PS were used to observe the typical MMAF-related irregular flagella morphological phenotypes, TEM was used to further inspect the sperm flagellum defects in ultrastructure, and IF was chosen to confirm the location of protein. Our study suggests that LRRC46 is an essential protein for sperm flagellum biogenesis, and its mutations might be associated with MMAF that causes male infertility. Thus, our study provides insights for understanding developmental processes underlying sperm flagellum formation and contribute to further observe the pathogenic genes that cause male infertility.

## 1. Introduction

Human infertility, due to a wide range of heterogeneous phenotypes, is defined as being unable to achieve a clinical pregnancy despite one year of regular and unprotected intercourse [1]. As a widespread health issue, infertility affects 10–15% of couples [2]. Causing about 45–50% infertility cases, male infertility is clinically diagnosed as azoospermia, asthenozoospermia (lower motility of sperm), oligozoospermia (lower counts of sperm), teratozoospermia (higher proportion of defective spermatozoa with disrupted morphology), and combinations of these multiple defects [3,4,5,6,7,8]. MMAF is characterized as the combination of absent, short, coiled, bent, or irregular sperm flagellum [1,7,8,9]. Via TEM, the mutations of MMAF genes often present abnormalities of axonemes, the absence of outer dense fiber, dysplasia of fibrous sheath, or malformations of the mitochondria [10,11]. Genetic origin accounts for 35–60% of these MMAF cases. Several gene-families are identified as associated with flagella, including the AKAP family, DANI family, DNAH family, RSPH family, CCDC family, CFAP family, TTC family, and some single genes [1,9,12,13,14,15,16,17,18,19,20,21,22,23,24,25,26,27,28,29,30,31,32,33,34,35,36,37,38,39,40,41,42,43,44,45,46,47,48,49,50,51,52,53,54].

Leucine-rich repeat proteins (LRR), that universally exist in viruses to eukaryotes, consist of 2–45 motifs of 20–30 amino acids in length that generally folds into an arc or horseshoe shape [55,56,57]. Proteins containing LRRs are reported as tyrosine kinase receptors, cell-adhesion molecules, and virulence factors which function in most of biological processes, such as DNA repair, recombination, transcription, RNA processing, immune response, and cell adhesion [58,59]. The deletion of *Lrrc50* causes dynein arm defects, immotile cilia, and male infertility [60]. The mutation of *Lrrc6*, an important paralog of *Lrrc46*, causes 10.6% primary ciliary dyskinesia by the absence of dynein arm components [61,62]. Outer dynein arms intermediate chain DNAI2 and of the inner dynein arms component DNALI1 are absent in *Lrrc6* mutations, indicating that *Lrrc6* is essential for proper axonemal assembly of inner and outer dynein arms [62]. *Lrrc6* is confirmed to colocalize with the centriole markers including SAS6 and PCM1, and interact with some dynein arms including ZMYND10 and Reptin/Ruvbl2 [63,64]. Observing the high similarities in sequences with *Lrrc6* phenotypes and functions, we hypothesize *Lrrc46* may present similar biological functions and phenotypes. To further investigate the essential role of LRRC46 in the late phase of spermatogenesis, we analyze the impact of the loss of LRRC46 function on flagellum formation and male fertility via the CRISPR/Cas9-mediated genome editing system.

## 2. Results

### 2.1. LRRC46 Is Specifically Expressed in Adult Testes

Analysis of RNA profiling data in the Mouse ENCODE project and NCBI (https://www.ncbi.nlm.nih.gov/gene/69297) (accessed on 1 May 2021), indicated that *Lrrc46* mRNA is present at very high levels in adult testes, at low levels in adult ovaries and lungs, and at only baseline levels in other mouse organs. Corresponding to the expression analysis in the NCBI, the RT-PCR analysis of multiple organs show that *Lrrc46* is expressed abundantly in adult testis, at a low level in the adult ovary, and is not apparently expressed in other organs (Appendix A).

At the protein level, immunoblotting of LRRC46 showed expression in the adult testis while no expression in other organs, including the heart, liver, spleen, lung, kidney, or ovary (Figure 1a). A developmental time series analysis showed that that *Lrrc46* was initially detected in the testis at postnatal day 21 (PD21), with the level increasing continuously from postnatal PD25 onward to a plateau at PD36, indicating that Lrrc46 might function in spermatogenesis (Figure 1b).

### 2.2. Generation of Lrrc46 Mutant Mice

With an ATG start codon in exon 1 and TGA stop codon in exon 8, the 6.81 kb testis-specific *Lrrc46* gene is located on chromosome 11. To characterize the potential functions of *Lrrc46* during spermatogenesis, *Lrrc46^−/−^* mice were created using a CRISPR/Cas9-mediated genome editing system from Cyagen Biosciences. Selecting exon 3 to exon 7 as the target sites (Figure 1c), 4075 bp were deleted. Via genomic DNA sequencing, the founder animals were genotyped and confirmed the genome edit. The total size of the *Lrrc46* locus in *Lrrc46^+/+^* mice is 697 bp while the knockout mice was 398 bp (Figure 1d). Immunoblotting of testis extracts indicated that no LRRC46 protein was expressed in *Lrrc46^−/−^* mice (Figure 1e).

### 2.3. Lrrc46 KO Does Not Impact Testes Size but Causes Male Infertility

At the gross and then histological levels, examining the testis structure of adult male mice led to our identification of phenotypic changes in the *Lrrc46* knockout male mice. Compared to the wild-type (WT) mice, the in the *Lrrc46* knockout mice showed no significant differences in the testis size, testis weight, body weight, or the ratio of testes weight to mouse body weight (Figure 1f–i). We further examined the fertility of *Lrrc46* male and female knockout mice. Exhibiting normal mounting behaviors, the *Lrrc46* knockout male mice produced copulatory plugs after mating to WT adult female mice, but produced no pups (Figure 1j). In contrast, *Lrrc46* knockout female mice generated offspring after mating with similarly aged wild-type males (Appendix A). HE-staining of ovary sections revealed no differences in the number or morphology of follicles between *Lrrc46^+/+^* and *Lrrc46^−/−^* female mice (Appendix A). These results establish that *Lrrc46* is required for male fertility but has no effect in female fertility.

### 2.4. Lrrc46 Is Required for Spermatogenesis

HE-staining of testes sections revealed obvious phenotypes in the sperm within the seminiferous tubules of *Lrrc46^−/−^* mice. Briefly, the seminiferous tubules of the *Lrrc46^+/+^* mice had a tubular lumen with flagella appearing from the developing spermatids, while the flagella were absent in the seminiferous tubules of *Lrrc46^−/−^* mice (Figure 2a, red arrow). Subsequent IF-staining for α/β-tubulin—a specific flagellum marker—confirmed the defects in sperm flagellum biogenesis resulting from *Lrrc46^−/−^* male mice (Figure 2b). These observations clearly suggest that *Lrrc46* functions in flagellum formation.

To characterize the potential functions of *Lrrc46* during sperm development, we further performed PAS experiments. PAS analysis showed that all of the seminiferous epithelium components and stages could be detected in both the *Lrrc46^+/+^* and *Lrrc46^−/−^* male mice (Figure 2c). The most prominent defects were observed in the spermatids at all stages of spermatogenesis: abnormally elongated and constricted sperm head shapes were identified in the *Lrrc46^−/−^* male mice seminiferous tubules. Some head defected mature spermatozoa were found in the Stage IX in the *Lrrc46^−/−^* male mice seminiferous tubule, which indicated that the absence of *Lrrc46* influences the normal sperm release (Figure 2c).

We next analyzed the sperm head shaping between *Lrrc46^+/+^* and *Lrrc46^−/−^* male mice, and no differences in *Lrrc46^−/−^* materials from the Golgi apparatus period (step 1 to step 3) to the capping period (step 4 to step 8) were observed. During acrosomal migration (step 9 to step 13) and the chromatin condensation period (step 14 to step 16), we found major morphological differences in the sperm heads between the seminiferous tubules of *Lrrc46^+/+^* and *Lrrc46^−/−^* male mice. Interestingly, head shaping was initiated at step 9 to step 10, and the morphology of the elongated *Lrrc46^−/−^* spermatid heads was roughly normal compared with that of Lrrc46^+/+^mice. During step 11 to step 18 of the spermatids head shaping period, the sperm presented abnormal, club-shaped head morphology in *Lrrc46^−/−^* male mice while *Lrrc46^+/+^* mice presented normal, hook-shaped heads (Figure 2d). This phenomenon became apparent between step 11 and step 16, which indicated that the absence of *Lrrc46* defects the normal sperm head shaping [65]. Together, these results indicate that *Lrrc46* is required for normal spermatogenesis.

### 2.5. Knockout of Lrrc46 in Mice Leads to MMAF Phenotypes and Low Sperm Motility

HE-staining of caudal epididymides transverse sections revealed only a few spermatozoa in the epididymal lumen of *Lrrc46^−/−^* mice. In addition, some dead cells could be detected in the *Lrrc46^−/−^* seminiferous tubules (Figure 3a). Then, we examined the spermatozoa that were released from the caudal epididymis and found the sperm count in the *Lrrc46^−/−^* mice to be significantly decreased compared with the *Lrrc46^+/+^* mice (Figure 3b). To further observe the single sperm morphological characteristics by PS, we found that the *Lrrc46^−/−^* sperm exhibit MMAF phenotypes, which are short, coiled, irregular, or absent flagella (Figure 4a). The percentage of the short, coiled, irregular, or absent flagella in the *Lrrc46* knockout mice were higher than in the wild-type mice (Figure 4f–i). The ratio of spermatozoa with abnormal heads and flagella in the *Lrrc46^−/−^* mice is significantly higher than in the wild-type mice (Figure 4d,e). IF experiments of anti-α/β-tubulin (red) antibodies, lectin peanut agglutinin (PNA, green), and DAPI (blue) in single spermatozoa also indicated that *Lrrc46^−/−^* sperm exhibit MMAF phenotypes (Figure 4b). Using SEM to observe ultrastructure of sperm flagellum revealed that the *Lrrc46^−/−^* sperm had a variety of abnormal head shapes, as well as coiled, short, or absent flagella while the spermatozoa from wild-type mice had normal morphology (Figure 4c). Notably, we observed the mitochondrial sheath was defected in the midpiece of sperm flagellum (Figure 4a,c). IF analysis with Mito Tracker, which is used to visualize mitochondria, showed that the mitochondrial sheath was malformed in the *Lrrc46^−/−^* spermatozoa (Appendix A).

Via CASA, the *Lrrc46* knockout male mice presented a significantly decreased sperm motility rate compared to *Lrrc46^+/+^* mice. The percentage of motile spermatozoa of the *Lrrc46^+/+^* mice was 77.67% and the progressive spermatozoa of the *Lrrc46^+/+^* mice was 21.67% (*p* < 0.01), while no motile spermatozoa were found for the *Lrrc46^−/−^* mice (Figure 3c,d). We observed significant differences in sperm motility *Lrrc46^−/−^* mice VAP, VSL, and VCL compared to the WT mice (Figure 3e–g) (*p* < 0.01), as three important parameters to further assess spermatozoa motility. These results indicated that *Lrrc46* is important for maintaining normal sperm motility.

### 2.6. Loss of Lrrc46 Causes Sperm Flagellum Ultrastructure Abnormalities

Using TEM analysis, we further observed the sperm flagellum defects in *Lrrc46* knockout mice in ultrastructure. Cross sections of the *Lrrc46^+/+^* mice sperm samples revealed the components of the sperm head and flagellum, including the acrosome of the head, mitochondrial sheath, the outer dense fibers of the midpiece, and the fibrous sheath of the principal piece. A TEM of the *Lrrc46^+/+^* mice sperm samples showed sperm native architecture, which included nine obvious doublet microtubules and a central pair microtubule structure (9+2) [10,11]. Further observation of the ultrastructure of sperm flagellum revealed that *Lrrc46^−/−^* mice show hypertrophy and hyperplasia of mitochondria, disarrangement and hyperplasia of outer dense fibers, and an absence of the fibrous sheath, compared to doublet microtubules and a central pair microtubule structure (9+2) in the proper position in *Lrrc46^+/+^* mice sperm flagellum (Figure 5a). In the *Lrrc46^−/−^* flagella, the central pair microtubule structure was absent or disrupted, various peri-axonemal microtubule components were also distorted, and nine double microtubule doublets split with each other (Figure 5b). We observed detachment of the acrosome from the nuclear envelope from step 11 to step 16 in the *Lrrc46^−/−^* flagella (Appendix A), which indicates that the absence of *Lrrc46* led to acrosomal defects in sperm. These sperm flagellum morphology defects in ultrastructure confirmed that *Lrrc46* may function for sperm flagellum formation or maintaining sperm flagellum stability.

### 2.7. Abnormal Sperm Manchettes Morphology in Lrrc46^−/−^ Male Mice

Via the PAS (Figure 2c,d) and TEM (Figure 6), abnormally elongated and constricted sperm head shapes were observed in the *Lrrc46^−/−^* male mice seminiferous tubules. The presence of long and abnormal spermatid heads suggests defects in the function of the manchette, which is involved in sperm head shaping. In order to identify the prominent causes of abnormal head shaping in *Lrrc46^−/−^* male mice, we use the antibody against acetylated tubulin, a flagellum-specific marker. IF experiments of anti-α/β-tubulin antibodies, lectin peanut agglutinin (PNA), and DAPI in single spermatozoa showed that manchette formation was normal in step 8 spermatids in *Lrrc46^−/−^* mice, while step 9 to step 13 spermatids of *Lrrc46^−/−^* mice had abnormally long and deformed manchettes compared with the WT controls (Figure 7a). These data indicated that the ablation of *Lrrc46* caused abnormal spermatozoa manchette formation from step 8 to step 13. The manchette of *Lrrc46^−/−^* spermatozoa displayed abnormal elongation.

### 2.8. Lrrc46 Is Located at All Sperm Flagellum Essentially at the Midpiece

Ablation of *Lrrc46* affects the normal male spermatozoa head and flagellum, but how the LRRC46 protein works during sperm development is still unknown. We performed the IF of anti-*Lrrc46* (red) antibodies, α/β-tubulin (green), and DAPI (blue) in spermatozoa from the *Lrrc46^+/+^* and *Lrrc46^−/−^* male mice. The IF analysis shows that *Lrrc46* expressed very strongly at the midpiece of the flagellum, and lightly throughout the whole wild-type sperm flagellum, whereas α-tubulin marked the whole tail in the elongated spermatids around step 15 to step 16, suggesting *Lrrc46* an axonemal pattern (Figure 7b). No *Lrrc46* staining was detected in the *Lrrc46^−/−^* mice flagellum (Figure 7b). In the last phase of spermiogenesis, *Lrrc46* appears to be required in spermatid elongation, and development of the sperm flagellum.

## 3. Discussion

In the present study, our generation and characterization of *Lrrc46^−/−^* mice with deletion of exons 3–7 identified the essential role of *Lrrc46* in spermatogenesis and male fertility. *Lrrc46* is expressed strongly in mouse testes, and we demonstrate that *Lrrc46* knockout mice represent a male infertility model with characteristic MMAF manifestations, including short, coiled, irregular, and absent flagella; these knockout mice have reduced total sperm count and abnormalities in sperm motility. TEM analysis of testis cross sections revealed that the short flagella of *Lrrc46^−/−^* male mice are disordered at their fibrous sheaths, outer dense fibers, manchette, and mitochondria. Observations of the spermatozoa from *Lrrc46^−/−^* male mice clearly revealed the detachment of the acrosome from the nuclear envelope. IF experiments in *Lrrc46^+/+^* mice sperm indicated that *Lrrc46* is located at the flagellum midpiece, where it apparently functions in flagellum formation. Our study provides evidence that the dysregulation of *Lrrc46* can deleteriously affect sperm flagellum development and stability.

First introduced in 2014, the MMAF phenotype—which was proposed to establish a clear definition for a constellation of sperm phenotypes in infertility patients—is now understood as a major cause male infertility in couples [1,7,9]. MMAF flagella present ultrastructure abnormalities of axonemes, lack outer dense fibers or fibrous sheaths, and show hypertrophy and hyperplasia of mitochondria. Described as the severest phenotype for sperm morphological defects, MMAF sperm flagella are short, coiled, irregular, or even absent [7,8,10]. About 35–60% of MMAF infertility cases are associated with a genetic etiology, and MMAF has been reported to be associated with a variety of genes in humans and in mice [9]. In previous studies, several gene-families were identified to be associated with flagella, including the AKAP family, DANI family, DNAH family, RSPH family, CCDC family, CFAP family, TTC family, and some single genes [1,12,13,14,15,16,17,18,19,20,21,22,23,24,25,26,27,28,29,30,31,32,33,34,35,36,37,38,39,40,41,42,43]. In the AKAP family, AKAP3 and AKAP4 mutations were identified causing dysplasia of the fibrous sheath, presenting MMAF phenotypes and male infertility [12,14,15]. In the DANI family, DANI1 and DANI2 were reported as encoding the outer dynein arm of flagellum and resulted in male sterility [16,17]. In the DNAH family, DNAH1 and DNAH2 encoded the inner arm heavy chain dynein components, DNAH6 and DNAH8 encoded the dynein axonemal heavy-chain components, and DNAH11 and DNAH17 encoded the outer dynein arm complexes components [1,18,19,20,21,22,23]. Mutations of these DNAH genes caused structural defects of the axoneme, presented MMAF phenotypes, and male infertility. In the RSPH family, RSPH9 and RSPH4A encoded a generic component of the radial spoke heads of cilia, and mutation of these genes caused central microtubular pair abnormalities and infertility [24,25,26]. TTC21A encodes an intra-flagellar transport (IFT)-associated protein that is essential for cilia motility [39]. Encoding an evolutionarily conserved axonemal protein, TTC29 is essential for normal flagellar structure and motility [38], and mutation of both TTC21A and TTC29 caused male sterility due to MMAF [38,39]. Mutations in CCDC39, CCDC40, and CCDC103 presented cilia axonemal disorganization, absent inner dynein arms, and male infertility [27,28,29]. Several CFAP family genes were associated with cilia and flagella abnormalities in human and mutant mouse models, including CFAP43, CFAP44, CFAP58, CFAP61, CFAP65, CFAP69, CFAP70, and CFAP251, are associated with flagellum biogenesis and morphogenesis, when mutant, caused MMAF and male infertility [30,31,32,33,34,35,36,37]. Furthermore, whole-exome sequencing (WES) identified FSIP2, DZIP1, and QRICH2 were associated with MMAF, and defected the axoneme [40,41,42,43].

Previous studies have indicated *Lrrc46* was found as a novel candidate susceptibility gene in familial prostate cancer (PCa) [66], high-grade serous ovarian cancer (HGSOC) [67,68], Glioblastoma [69], and Nasopharyngeal carcinoma (NPC) [70]. In a silico data study, *Lrrc46* was first described in the testes, which is involved not only primarily in structure elaboration or maintenance of germ cell process, but also in acid nucleic metabolism [71]. In a presently prospective cohort study, they use array hybridization, quantitative real-time polymerase chain reaction, and Counter technology to find the specifically expressed genes that are associated with ectopic pregnancy (ECT) and miscarriage in women. They found five cilia-associated genes, including *C20orf85, Lrrc46, Rsph4a,* and *Wdr49* [72]. Most of the cilia-associated genes were predicted to regulate tubulin and microtubule assembly, or be a part of cilia complexes. The above studies indicated that *Lrrc46* was a cilia-associated gene. However, these studies did not give any analysis about *Lrrc46* function in severe tumors at all. Our results identified an essential role for *Lrrc46* in spermatogenesis and male fertility and show that *Lrrc46* deficiency leads to an atypical (relatively severe) MMAF phenotype, thus improving our understanding of the mechanisms underlying sperm flagellum development and head shape during spermiogenesis. These findings can facilitate the development of improved treatments for gene-affected male infertile individuals.

## 4. Materials and Methods

### 4.1. Generation of Lrrc46 Knockout Mice via CRISPR/Cas9 Gene Editing System

The mouse *Lrrc46* gene (Transcript: ENSMUSG00000020878) is 6.81 kb and located on chromosome 11, containing 8 exons, with an ATG start codon in exon 1 and a TGA stop codon in exon 8. Selecting exon 3 to exon 7 as the target site, *Lrrc46* knockout mice were created using a CRISPR/Cas9-mediated genome editing system from Cyagen Biosciences. The gRNA and Cas9 mRNA were co-injected into fertilized eggs of C57BL/6 mice, to generate a targeted line with a 4075 bp base deletion, CGTCAGGAACGTCCTAGAAGAGG-del-4075bp AATCTCACTGACACCGTCCTAGG. The founder animals were identified by DNA sequence analysis. Intercross heterozygous-targeted mice were created to generate normal offspring. Using PCR amplification of genomic DNA that was extracted from mouse tails, all the mice’s genotypes were distinguished. For the *Lrrc46* knockout mice, the specific primers were forward: 5-TGGTATAATCCTGGCCCTCAAG-3 and reverse: 5-GTTCCTGATGCTGATGCAGTTC-3, yielding a 697 bp fragment. Similarly, forward:5-GCAGGACTGTGTTCAAGGCGGTG-3 and reverse: 5-TGGTATAATCCTGGCCCTCAAG-3 were performed for the *Lrrc46* wild-type mice, yielding a 398 bp fragment. The PCR annealing temperature was 60.0 °C. The mice and animal care protocols were approved by the Regional Ethics Committee of the School of Medicine, Shandong University.

### 4.2. Tissue Collection and Histological Analysis

*Lrrc46^+/+^* and *Lrrc46^−/−^* male mice (n = 5) for each genotype were used in this experiment. After being killed by euthanasia, all the samples were dissected, fixed in 4% paraformaldehyde (P1110, Solarbio, Beijing, China) or Bouin’s for 24 h at 4 °C. After being dehydrated in 70% ethanol, all the samples were embedded in paraffin, then sectioned at 5 μm and mounted on glass slides for staining. The slides were then stained with hematoxylin and eosin (HE) and PAS staining routinely.

### 4.3. RNA Extraction, cDNA Synthesis, and RT-PCR

For the RT-PCR experiments to analyze the expression of *Lrrc46* mRNAs in various organs, all the RNA was isolated from *Lrrc46* wild-type adult mice organs, including heart, liver, spleen, kidney, colon, muscle, fat, brain ovary, and testis. All the cDNA samples were synthesized using the Prime Script TM RT Reagent Kit with gDNA Eraser and TB Green Premix Ex Taq (Takara, RR037A). All RT-PCR reaction experiments were started with an initial denaturation at 95 °C for 10 min, followed by 30 cycles of denaturation at 95 °C for 30 s, then annealed at 65 °C for 30 s, extension at 72 °C for 1 min, and a final extension at 72 °C for 10 min using a T100 Thermal Cycler (Bio-Rad). The specific forward and reverse primers were as followed: *Lrrc46* primer Forward: 5-TCACACTCTCGGTGAACTGG-3 and Reverse: 5-GCTCCTATCACCATCTTCCTGT-3. β-actin was amplified as a housekeeping gene. β-actin primer Forward: 5-CATTGCTGACAGGATGCAGAAGG-3 and Reverse: 5-TGCTGGAAGGTGGACAGTGAGG-3, yielding a 351 bp fragment.

### 4.4. Immunoblotting

To prepare the tissue proteins, the tissues were extracted using lysis buffer (P0013, Biyuntian, Shanghai, China) and 1% (*v/v*) protease inhibitor cocktail (4693116001, Roche, Shanghai, China). After tissue grinding and ultrasonication, the samples were incubated on ice for 25 min. Then, the protein lysates were centrifuged at 12,000 rpm for 25 min at 4 °C. The supernatant liquid of the extracts was used for immunoblotting. The proteins were separated by SDS-PAGE and electro-transferred onto a nitrocellulose membrane. Then, the membrane was blocked in 5% skim milk overnight and then incubated with corresponding primary and secondary antibodies.

### 4.5. Mouse Sperm Collection

The caudal epididymides were surgically and immediately dissected from the *Lrrc46^+/+^* and *Lrrc46^−/−^* male mice. They were squeezed out from the caudal epididymis, the spermatozoa were released in 1 mL phosphate buffered saline (PBS) for 30 min at 37 °C, under 5% CO_2_ atmosphere. Then the samples were provided for sperm counting, morphological analysis, and IF experiments.

### 4.6. Single Sperm Morphological Analysis

The sperm morphological analysis was evaluated accorded to the World Health Organization (WHO) guidelines. From every *Lrrc46* wild-type and knockout mouse, more than 250 spermatozoa were analyzed. A total of 10 μL sperm samples from each mouse was mounted on glass slides and dried overnight. Then these glass slides were performed PS. Diluted at a ratio of 1:100, via hemocytometer all the sperm samples were counted at least five times.

### 4.7. Sperm Motility Assessment Using CASA

*Lrrc46^+/+^* and *Lrrc46^−/−^* male mice (n = 10) for each genotype were killed by euthanasia. The cauda epididymis was quickly cut into pieces in 1 mL human tubal fluid (HTF) (Millipore) or PBS, then incubated for 5 min at 37 °C. After incubating, 10 μL sperm liquid taken from each sample was put on the glass slide (Hamilton Thorne, 80 micron 2X-CEl), and taken captures via a CCD camera. Via a computer-assisted animal motility system (Version.12 CEROS, Hamilton Thorne, Beverly, MA, USA), more than 250 spermatozoa from every sample were performed by computer-associated semen analysis (CASA), using an Olympus B x51 microscope through a 20X phase objective (Olympus, Tokyo, Japan). Via computer-associated semen analysis (CASA), the sperm samples were observed to evaluate the parameters of total sperm motility, including the motile spermatozoa, progressive spermatozoa, average path velocity (VAP), progressive velocity (VSL), the track speed (VCL), and so on.

### 4.8. Scanning Electron Microscopy (SEM)

*Lrrc46^+/+^* and *Lrrc46^−/−^* male mice (n = 3) for each genotype were killed by euthanasia. The cauda epididymis was quickly cut into pieces in 1 mL PBS. After shredding, 10 μL sperm liquid taken from each sample was put on the coverslips. Fixed with 2.5% glutaraldehyde, all coverslips were post-fixed in osmium tetroxide, then washed with water for at least for three times, dehydrated through 50%, 70%, 95%, and 100% ethanol. Via Tousimis Autosamdri-810 Critical Point Dryer, the coverslips were dried at 55 °C, a critical point. The samples then mounted onto specimen stubs, these coverslips were sputter-coated with palladium, and viewed with a FEI Quanta ESEM 200 SEM.

### 4.9. Transmission Electron Microscopy Analysis (TEM)

*Lrrc46^+/+^* and *Lrrc46^−/−^* male mice (n = 5) for each genotype were killed by euthanasia. All the adult testis and epididymis samples were cut into pieces of approximately 1 mm^3^, and fixed immediately in 2.5% (*v/v*) glutaraldehyde overnight for 4 °C. All the samples were immersed in 1% OsO4 for 1 h at 4 °C, gradually dehydrated using a series of acetone, and then embedded in 100% resin. Then, cutting on an ultramicrotome, the samples ultrathin sections were stained with uranyl acetate and lead citrate. Lastly, the sections were imaged and analyzed via a transmission electron microscope (JEM-1400, JEOL, Tokyo, Japan).

### 4.10. Antibodies

Used for immunoblotting, IF, the primary antibodies included rabbit anti-LRRC46 polyclonal antibody (1:200 dilution, NBP2-62692, NOVUS, Taiwan, China), mouse anti-acetylated-a-tubulin monoclonal antibody (1:2000 dilution, T7451, Sigma-Aldrich, DEDAR, GER). Meanwhile, Alexa Fluor 488-, 594-conjugated secondary antibodies were used for detection with the primary antibodies. The secondary antibodies were Alexa Fluor goat anti-rabbit 488 (1:200 dilution, ab150077, Abcam, MA, USA), Alexa Fluor goat anti-rabbit 594 (1:200 dilution, ab150080, Abcam, MA, USA), Alexa Fluor goat anti-mouse 488 (1:200 dilution, ab150113, Abcam, MA, USA), Alexa Fluor goat anti-mouse 594 (1:200 dilution, ab150116, Abcam, MA, USA), and Alexa Fluor 488 conjugate of lectin PNA 351 (1:1000 dilution, L21409, Thermo Fisher Scientific, Waltham, MA, USA), Mito Tracker Deep Red 633 (1:2000 dilution, M22426, Thermo Fisher Scientific, Waltham, MA, USA).

### 4.11. IF in Testes

*Lrrc46^+/+^* and *Lrrc46^−/−^* male mice (n = 6) for each genotype were used in this experiment. After being killed by euthanasia, all the samples were dissected, fixed in 4% paraformaldehyde (P1110, Solarbio, Beijing, China) or Bouin’s for 24 h at 4 °C. After being dehydrated in 70% ethanol, all the samples were embedded in paraffin, then sectioned at 5 μm and mounted on glass slides. These sections were deparaffinized in 100% (*v/v*) dimethylbenzene for 30 min, rehydrated in 100%, 95%, 80%, 75% (*v/v*) ethanol dimethylbenzene for 25 min, antigen retrieval in 10 mM sodium citrate buffer (pH 7.2) for 25 min, washed several times in PBS (pH 7.4), followed by a decrease to room temperature. After being dropped to room temperature, these sections were blocked with 5%BSA for 1 h. After blocking with 5% BSA or 5% bovine serum albumin for 1 h, these sections were perforated with 0.5% (*v/v*) Triton-X 100. Then the primary antibodies were added to the glass slides and incubated at 4 °C for 14–16 h. These were washed several times in PBS (pH 7.4), the secondary antibodies were added for 1.5 h at 37 °C, the DAPI were added for 10 min 37 °C. All the IF samples were imaged by a confocal microscopy which contained the Leica Andor Dragonfly spinning disc confocal microscope driven by Fusion Software. The images were performed with ImageJ Software (NIH, version 1.6.0-65) or Bitplane Imaris (version 8.1) software.

### 4.12. IF in Single Spermatozoa

*Lrrc46^+/+^* and *Lrrc46^−/−^* male mice (n = 6) for each genotype were killed by euthanasia. The cauda epididymis was quickly cut into pieces in 1 mL PBS. A total of 10 μL mice sperm liquid that was taken from each sample was put on the glass slides and air-dried, then stored at −80 °C. All the sections were fixed in 4% PFA for 15 min, blocked with 5%BSA for 1 h at room temperature. After being blocked with 5%BSA or 5% bovine serum albumin for 1 h, these sections were perforated with 0.5% (*v/v*) Triton-X 100. Then the primary antibodies were added to the glass slides and incubated at 4 °C for 14–16 h. After being washed several times in PBS (pH 7.4), the secondary antibodies were added for 1.5 h at 37 °C, the DAPI were added for 10 min 37 °C.

### 4.13. IF in Testicular Germ Cells

*Lrrc46^+/+^* and *Lrrc46^−/−^* male mice (n = 6) for each genotype were killed by euthanasia. All the testes were immediately fixed with 2% paraformaldehyde in 0.05% PBST (PBS with 0.05% Triton X-100) or 5 min. About 20 μm fixed seminiferous segment tubules were placed on the glass slides that contained 100 μL fixation solution, covered with coverslips and were pressed down on. Then these glass slides were frozen in liquid nitrogen and stored at stored at −80 °C. After the coverslips were removed, these slides were fixed in 4% PFA for 15 min at room temperature and blocked with 5%BSA for 1 h. After being blocked with 5%BSA or 5% bovine serum albumin for 1 h, these sections were perforated with 0.5% (*v/v*) Triton-X 100. Then the primary antibodies were added to the glass slides and incubated at 4 °C for 14–16 h. After being washed several times in PBS (pH 7.4), the secondary antibodies were added for 1.5 h at 37 °C, the DAPI were added for 10 min 37 °C.

## 5. Statistical Analysis

In this paper, all of the experiments were repeated at least three times, and all of the presented error bars were denoted as the mean ± SEM. The statistical analyses were performed using GraphPad Prism (version 8.02) by the Student’s *t*-test with a paired, two-tailed distribution. The differences were considered significant while the *p* values < 0.05 (*), 0.01 (**), or 0.001 (***).

## Figures and Tables

**Figure 1 ijms-23-08525-f001:**
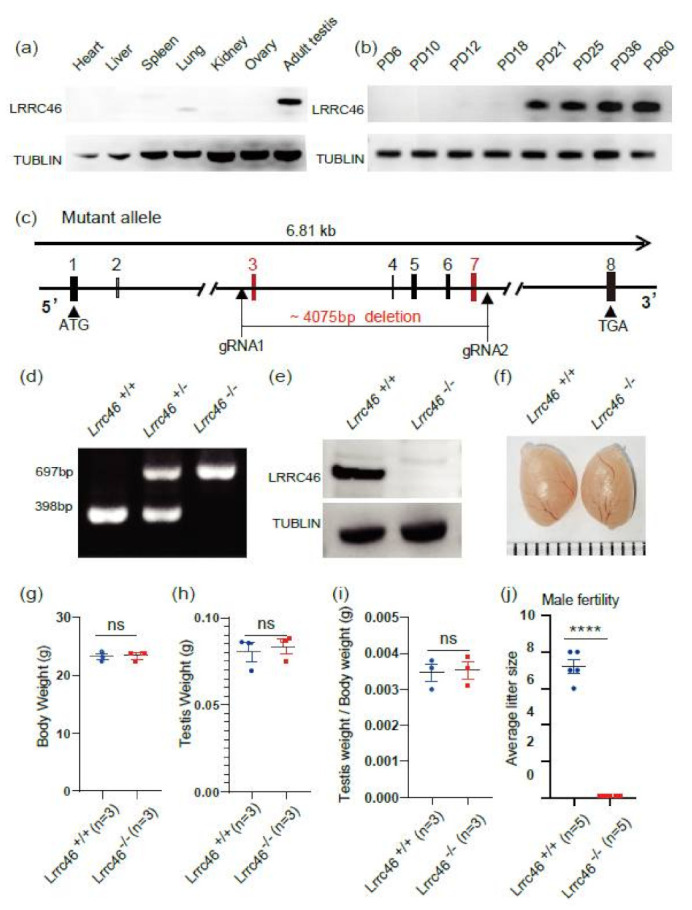
**The generation of *Lrrc46* knockout mice**. (**a**) With tubulin serving as an expression control, immunoblotting of *Lrrc46* is specifically expressed in adult testis, but there was low expression in lung and no expression in other organs, including heart, liver, spleen, kidney, or ovary. (**b**) With tubulin serving as an expression control, *Lrrc46* was initially expressed starting at postnatal day 21 in testis. (**c**) The *Lrrc46* knockout strategy in mice. (**d**) Genotyping identification of *Lrrc46* knockout mice. (**e**) Immunoblotting of LRRC46 in *Lrrc46^+/+^* and *Lrrc46^−/−^* testes. Tubulin was used as the control. (**f**) The size of the testis was comparable between *Lrrc46^+/+^* and *Lrrc46^−/−^* mice. (**g**) The body weights of *Lrrc46^+/+^* and *Lrrc46^−/−^*male mice (n = 3 independent experiments); (**h**) The testis weights of *Lrrc46^+/+^* and *Lrrc46^−/−^* male mice (n = 3 independent experiments); (**i**) The ratio of testis weight to body weight in the *Lrrc46^+/+^* and *Lrrc46^−/−^* male mice (n = 3 independent experiments); (**j**) The average litter size of *Lrrc46^+/+^* and *Lrrc46^−/−^* male mice in 3 months (n = 5 independent experiments). KO males were completely sterile. Data are presented as the mean ± SD. **** *p* < 0.0001, ns (no significant differences).

**Figure 2 ijms-23-08525-f002:**
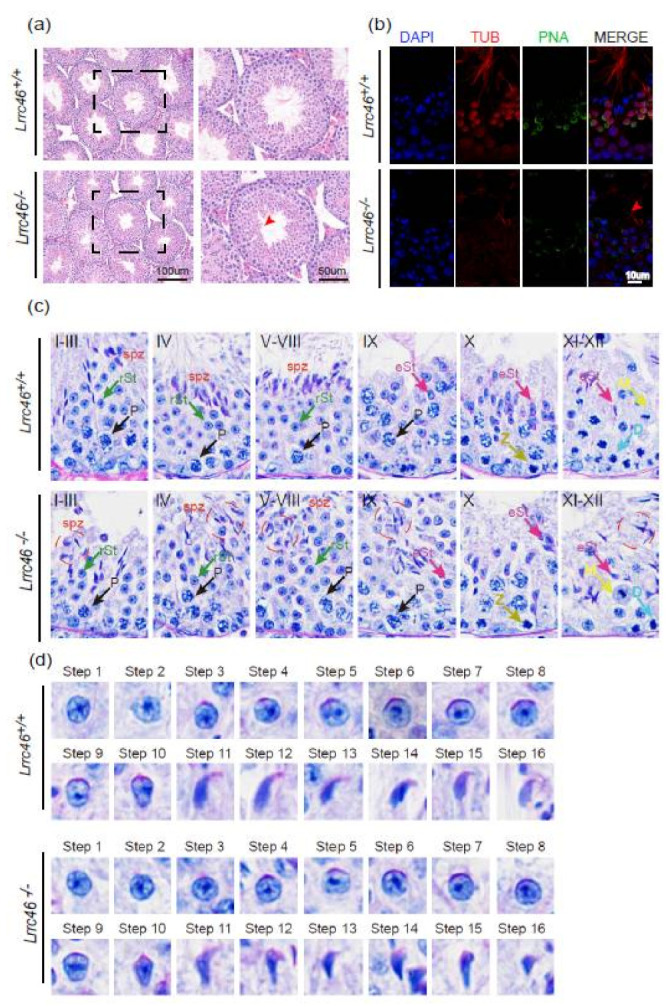
**Spermatogenesis defects of *Lrrc46* knockout mice.** (**a**) HE-staining of testes sections from the *Lrrc46^+/+^* and *Lrrc46^−/−^* male mice. (**b**) IF of anti-α/β-tubulin (red) antibodies in testes sections from the *Lrrc46^+/+^* and *Lrrc46^−/−^* male mice. (**c**) The PAS and HE-staining analysis of the testis seminiferous tubule cross-sections of *Lrrc46^+/+^* and *Lrrc46^−/−^* male mice. The arrows highlight germ cells at various stages of spermatogenesis. Defects in the nuclear shape of several elongating spermatids were clearly evident in the *Lrrc46^−/−^* male mice seminiferous tubule (red circle). P: pachytene spermatocyte, D: diploneme spermatocyte, Z: zygotene spermatocyte, M: meiotic spermatocyte, rST: round spermatid, eST: elongating spermatid, spz: spermatozoa. (**d**) The PAS and HE-staining analysis of spermatids at different steps from *Lrrc46^+/+^* and *Lrrc46^−/−^* male mice. During step 1 to step 10 spermatids of acrosome development period, the sperm acrosome morphology was roughly normal in the *Lrrc46^−/−^* male mice. During step 11 to step 18, te spermatids head shaping period, the sperm had an abnormal, club-shaped sperm head morphology in *Lrrc46^−/−^* male mice while *Lrrc46^+/+^* mice had normal, hook shaped heads.

**Figure 3 ijms-23-08525-f003:**
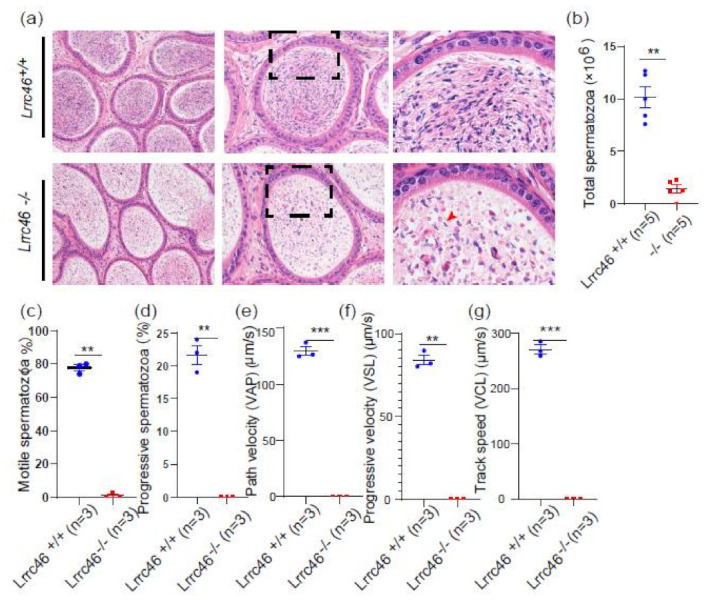
**Knockout of *Lrrc46* presents low sperm counts and motility.** (**a**) HE-staining of the caudal epididymis from the *Lrrc46^+/+^* and *Lrrc46^−/−^* male mice. Apoptotic bodies were detected in *Lrrc46^−/−^* caudal epididymis sections (red arrowheads). (**b**) The sperm counts in the caudal epididymis were significantly decreased in the *Lrrc46^−/−^* male mice (n = 5 independent experiments). The data are presented as the mean ± SD. ** *p* < 0.01. (**c**,**d**). The percentage of motile spermatozoa (**c**), and progressive spermatozoa (**d**) in the *Lrrc46^+/+^* and *Lrrc46^−/−^* male mice (n = 3 independent experiments). The data are presented as the mean ± SD. ** *p* < 0.01. (**e**–**g**) The VAP (**e**), VSL (**f**), and VCL (**g**) of the spermatozoa in the *Lrrc46^+/+^* and *Lrrc46^−/−^*male mice (n = 3 independent experiments). The data are presented as the mean ± SD. ** or ***, *p* < 0.01 or *p* < 0.001.

**Figure 4 ijms-23-08525-f004:**
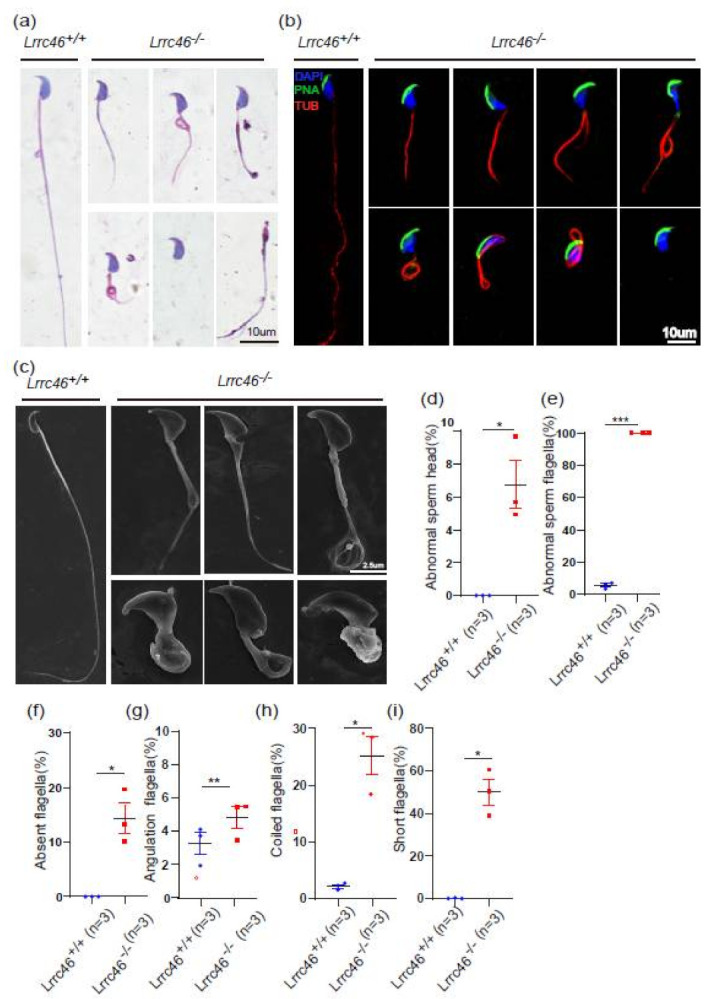
**Absence of *Lrcc46* results in MMAF.** (**a**) PS of spermatozoa from *Lrrc46^+/+^* and *Lrrc46^−/−^* male mice caudal epididymis, indicating abnormal spermatozoa such as abnormal head and coiled, short, or absent flagella. (**b**) IF of anti-α/β-tubulin (red) antibodies, PNA (green), and DAPI (blue) in spermatozoa from the *Lrrc46^+/+^* and *Lrrc46^−/−^* male mice, indicating abnormal spermatozoa such as abnormal head and coiled, short, or absent flagella. (**c**) Via SEM, spermatozoa from *Lrrc46^−/−^* male mice show severe flagella and head morphology defects. These abnormalities include a variety of defects including abnormal head and coiled, short, or absent flagella while spermatozoa from wild-type mice had normal morphology. (**d**) The percentage of abnormal spermatozoa head from *Lrrc46^+/+^* and *Lrrc46^−/−^* male mice caudal epididymis (n = 3 independent experiments). The data are presented as the mean ± SD. * *p* < 0.01. (**e**) The percentage of abnormal spermatozoa flagella from Lrrc46*^+/+^* and Lrrc46*^−/−^*male mice caudal epididymis (n = 3 independent experiments). The data are presented as the mean ± SD. *** *p* < 0.001. (**f**–**i**) The percentage of absent spermatozoa flagella (**f**), angulation spermatozoa flagella (**g**), coiled spermatozoa flagella (**h**), and short spermatozoa flagella (**i**). The data are presented as the mean ± SD. * or ** *p* < 0.01.

**Figure 5 ijms-23-08525-f005:**
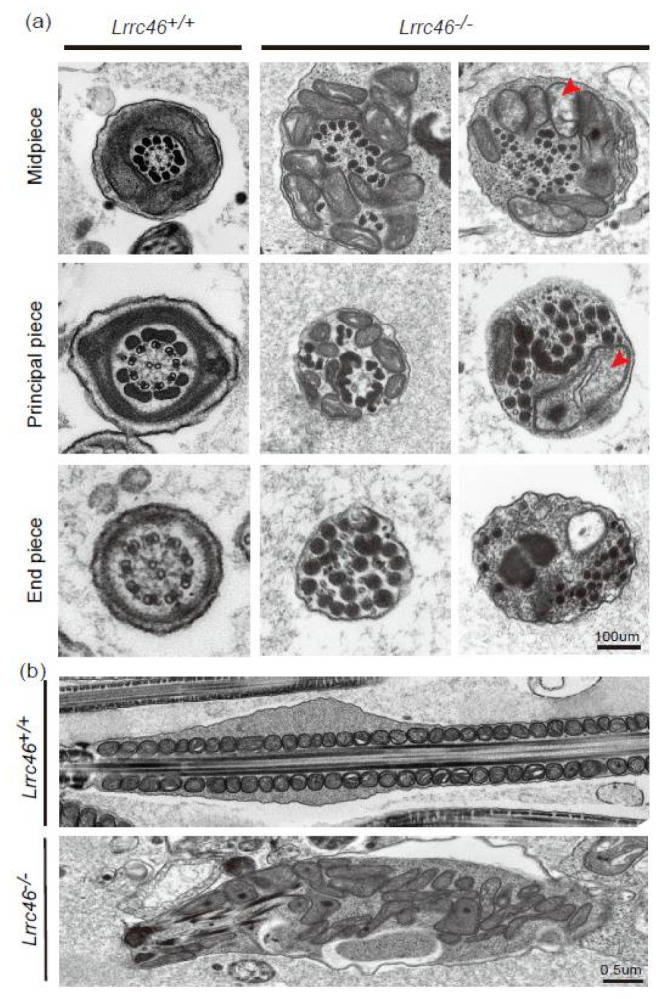
**Sperm tail ultrastructure are disorganized in *Lrrc46* KO mice.** (**a**) The ultrastructure of testes cross sections by TEM in the *Lrrc46^+/+^* and *Lrrc46^−/−^* male mice. Lacking the central pair of microtubules or 9+2 microtubule, most spermatozoa from *Lrrc46^−/−^* male mice presented discorded peripheral microtubules and hyperplasia of fibrous sheaths. Scale bar: 100 μm. (**b**) Via TEM, the spermatozoa from *Lrrc46^−/−^* male mice showed a severe axonemal disorganization. Scale bar: 0.5 μm.

**Figure 6 ijms-23-08525-f006:**
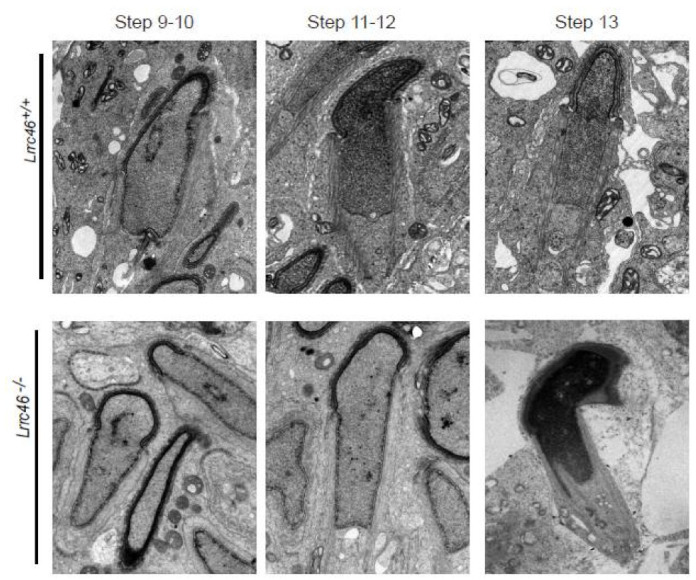
**Ablation of *Lrrc46* affects the normal spermatozoa manchettes morphology****in mice.** The ultrastructure of testes cross sections by TEM in the *Lrrc46^+/+^* and *Lrrc46^−/−^* male mice. TEM images of *Lrrc46^−/−^* step 9–13 spermatids showing the perinuclear ring constricting the sperm nucleus and causing abnormal sperm head formation. White arrows indicate the manchette microtubules. Scale bar: 100 μm.

**Figure 7 ijms-23-08525-f007:**
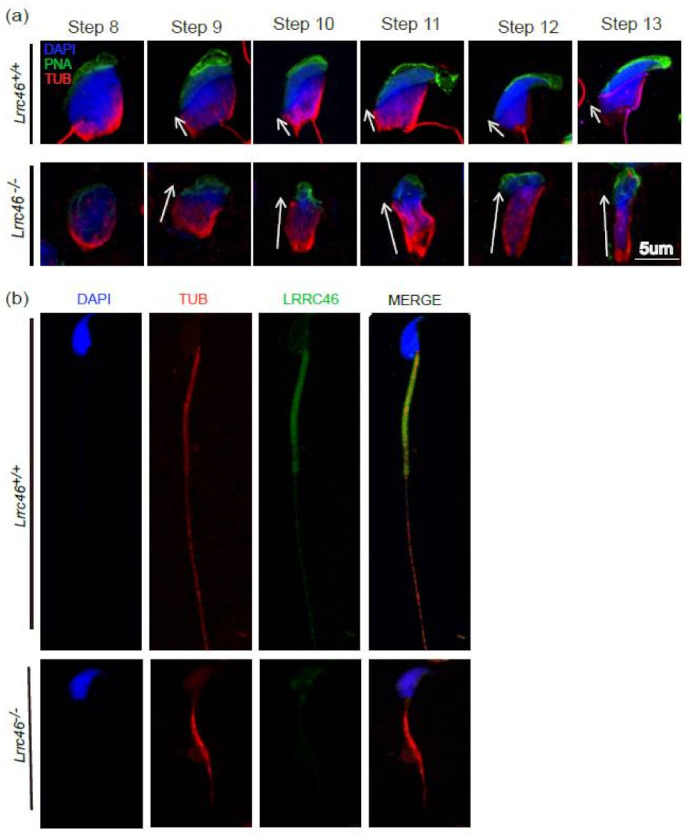
***Lrrc46* immunostaining of spermatozoa in mouse.** (**a**) IF of anti-α/β-tubulin (red), antibodies, PNA (green), and DAPI (blue) in spermatozoa from the *Lrrc46^+/+^* and *Lrrc46^−/−^*male mice, indicating the ablation of *Lrrc46* caused abnormal spermatozoa manchette formation from step 8 to step 13. The manchette of *Lrrc46^−/−^* spermatozoa displayed abnormal elongation. (**b**) IF of anti-α/β-tubulin (red) antibodies, LRRC46 (green), and DAPI (blue) in spermatozoa from the *Lrrc46^+/+^* and *Lrrc46^−/−^* male mice. *Lrrc46* located at all the sperm specifically at the midpiece in wild-type mice, while the *Lrrc46^−/−^* spermatozoa lost the normal location.

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
