# Peer review of "LRRC46 Accumulates at the Midpiece of Sperm Flagella and Is Essential for Spermiogenesis and Male Fertility in Mouse"

_ijms, 2022, doi:10.3390/ijms23158525_

Round 1
Reviewer 1 Report
Regarding manuscript entitled “LRRC46 accumulates at the midpiece of sperm flagella and is essential for spermiogenesis and male fertility in mouse”, the authors aimed to assess the role of LRRC46 in spermiogenesis and male fertility in mouse.
The study is interesting and well designed. However, minor revision needs before its acceptance.
Add clear aim to abstract and at the end of introduction section.
Abstract has nothing related to material and method. Describe the important part of material and method in abstract.
Clear conclusion must be added at the end of abstract.
“At least five Lrrc46+/+ and Lrrc46 124 -/- male mice” please be accurate and add the exact number of mice or samples used for each test (n=??).
Why the authors did not evaluate plasma membrane integrity and some other valuable parameters in sperm which can affect motility and fertility.
It is better to update your citations to 2022.
Author Response
Point-by-point responses of reviewers’ comment
Editor(s)' Comments to Author: Regarding manuscript entitled “LRRC46 accumulates at the midpiece of sperm flagella and is essential for spermiogenesis and male fertility in mouse”, the authors aimed to assess the role of LRRC46 in spermiogenesis and male fertility in mouse. The study is interesting and well designed. However, minor revision needs before its acceptance.
Authors’ response: We thank and appreciate the very constructive comments from the two reviewers as well as the editor. As directed, we have added additional aim in abstract and introduction as guided by the reviewers. We provide our full point-by-point responses to the reviewer comment, below. We trust that the reviewers and the editor will agree that the review process has here worked as designed. We hope that our revised manuscript will be deemed suitable for publication in International Journal of Molecular Sciences. Let us take this opportunity to again thank the reviewers and editor for their continued help our behalf.
Reviewer 1
Major comments:
- Comments: Add clear aim to abstract and at the end of introduction section.
Authors’ response: We appreciate the reviewer’s careful reading of our manuscript. We have added the aim to abstract in Line 45-46, and added the aim at the end of introduction in Line 98-101.
- Comments: Abstract has nothing related to material and method. Describe the important part of material and method in abstract.
Authors’ response: Thanks for bringing this omission to our attention, which we have now addressed as guided in Line 57-60.
- Comments: Clear conclusion must be added at the end of abstract.
Authors’ response: Thank you for focusing our attention on this issue. We have added the conclusion at the end of abstract in Line 60-64.
- Comments: “At least five Lrrc46+/+ and Lrrc46 124 -/- male mice” please be accurate and add the exact number of mice or samples used for each test (n=??).
Authors’ response: We sincerely apologize of this error. Now we have revised this as directed; thanks.
- Comments: Why the authors did not evaluate plasma membrane integrity and some other valuable parameters in sperm which can affect motility and fertility.
Authors’ response: Thank you for focusing our attention on this issue. Via computer-associated semen analysis (CASA), the sperm samples were observed the parameters of total sperm motility, including the motile spermatozoa, progressive spermatozoa, average path velocity (VAP), progressive velocity (VSL), the track speed (VCL), and so on. From amongst the three kinematic parameters, VCL, VSL, VAP, VAP shows the highest correlation with fertility, and it may be the most useful sperm speed/velocity parameter, which can be relied upon for the estimation of sperm fertility [1-2]. The paper to assess bull fertility and sperm cell velocity by CASA, they conclude that “The data of the experiments are summarized mainly with a focus on the effects of individual velocities (curvilinear velocity: VCL, straight-line velocity: VSL, average path velocity: VAP) on fertility rather than on the influence of progressive motility as a whole” [2]. And as descript in Fig 3c-3g, no motile spermatozoa were found for the Lrrc46-/- mice.
- Rijsselaere T, Van Soom A, Maes D, de Kruif A. Effect of technical settings on canine semen motility parameters measured by the Hamilton-Thorne analyzer. Theriogenology. 2003;60:1553–68. doi: 10.1016/S0093-691X(03)00171-7.
2.Nagy Á, Polichronopoulos T, Gáspárdy A, Solti L, Cseh S. Correlation between bull fertility and sperm cell velocity parameters generated by computer-assisted semen analysis. Acta Vet Hung. 2015;63:370–81. doi: 10.1556/004.2015.035.
- Comments: It is better to update your citations to 2022.
Authors’ response: We have revised this as directed; thanks.
Let us again take this opportunity to thank Reviewer #1 for the helpful comments, which have obviously improved the clarity and purport of our message.

Reviewer 2 Report
Yin et al. presented an animal study regarding the role of LRRC46 in biogenesis of sperm flagellum using wild type and male KO mice. Except for some grammar mistakes, the paper is well written, the methodology of study is appropriate and the results are scientifically relevant and clearly presented.
I have only some minor comments:
- Attention to the use of abbreviations. They are often used in the wrong way, while they should be presented in the extended form only the first time they are used.
- In the introduction, all the verbs should be at present (e.g., line 3 "infertility AFFECTS" rather than "infertility AFFECTED"
Author Response
Point-by-point responses of reviewers’ comment
Editor(s)' Comments to Author: Regarding manuscript entitled “LRRC46 accumulates at the midpiece of sperm flagella and is essential for spermiogenesis and male fertility in mouse”, the authors aimed to assess the role of LRRC46 in spermiogenesis and male fertility in mouse. The study is interesting and well designed. However, minor revision needs before its acceptance.
Authors’ response: We thank and appreciate the very constructive comments from the two reviewers as well as the editor. As directed, we have added additional aim in abstract and introduction as guided by the reviewers. We provide our full point-by-point responses to the reviewer comment, below. We trust that the reviewers and the editor will agree that the review process has here worked as designed. We hope that our revised manuscript will be deemed suitable for publication in International Journal of Molecular Sciences. Let us take this opportunity to again thank the reviewers and editor for their continued help our behalf.
Reviewer 2
Comments to the Author:Yin et al. presented an animal study regarding the role of LRRC46 in biogenesis of sperm flagellum using wild type and male KO mice. Except for some grammar mistakes, the paper is well written, the methodology of study is appropriate and the results are scientifically relevant and clearly presented.
Authors’ response: We thank the reviewer very much for his/her appreciation of our work, and we appreciate the very constructive comments and suggestions that have helped us to prepare what we feel is an obviously improved revised manuscript describing our study.
I have only some minor comments:
- Comments: Attention to the use of abbreviations. They are often used in the wrong way, while they should be presented in the extended form only the first time they are used.
Authors’ response: Thanks for bringing this error to our attention; we have corrected this in the revised manuscript.
- Comments: In the introduction, all the verbs should be at present (e.g., line 3 "infertility AFFECTS" rather than "infertility AFFECTED"
Authors’ response: We appreciate the reviewer’s careful reading of our manuscript. We have revised this as directed; thanks.
Let us take this opportunity to thank reviewer #2 for the extremely helpful comments and guidance, which has obviously improved the technical rigor and clarity of message in our study.
